# Zero-field nuclear magnetic resonance of chemically exchanging systems

Danila A. Barskiy [1,2], Michael C. D. Tayler [3,9], Irene Marco-Rius [4,10], John Kurhanewicz[4], Daniel B. Vigneron[4], Sevil Cikrikci[1,5], Ayca Aydogdu[1,5], Moritz Reh[6], Andrey N. Pravdivtsev[7], Jan-Bernd Hövener[7], John W. Blanchard [8], Teng Wu [8], Dmitry Budker[6,8] & Alexander Pines [1,2]

Zero- to ultralow-field (ZULF) nuclear magnetic resonance (NMR) is an emerging tool for precision chemical analysis. In this work, we study dynamic processes and investigate the influence of chemical exchange on ZULF NMR $J$-spectra. We develop a computational approach that allows quantitative calculation of $J$-spectra in the presence of chemical exchange and apply it to study aqueous solutions of [$^{15}$N]ammonium ($^{15}$NH$_4^+$) as a model system. We show that pH-dependent chemical exchange substantially affects the $J$-spectra and, in some cases, can lead to degradation and complete disappearance of the spectral features. To demonstrate potential applications of ZULF NMR for chemistry and biomedicine, we show a ZULF NMR spectrum of [2-$^{13}$C]pyruvic acid hyperpolarized via dissolution dynamic nuclear polarization (dDNP). We foresee applications of affordable and scalable ZULF NMR coupled with hyperpolarization to study chemical exchange phenomena in vivo and in situations where high-field NMR detection is not possible to implement.

[1] Department of Chemistry, University of California—Berkeley, Berkeley, CA 94720-3220, USA. [2] Materials Science Division, Lawrence Berkeley National Laboratory, Berkeley, CA 94720-3220, USA. [3] Department of Chemical Engineering and Biotechnology, University of Cambridge, West Cambridge CB3 0AS, UK. [4] Department of Radiology and Biomedical Imaging, University of California—San Francisco, San Francisco, CA 94158-2330, USA. [5] Food Engineering Department, Middle East Technical University, 06800 Ankara, Turkey. [6] Department of Physics, University of California—Berkeley, Berkeley, CA 94720-7300, USA. [7] Section for Biomedical Imaging, Molecular Imaging North Competence Center (MOIN CC), Department of Radiology and Neuroradiology, University Medical Center Schleswig-Holstein (UKSH), Kiel University, Am Botanischen Garten 14, 24118 Kiel, Germany. [8] Helmholtz-Institut Mainz, Johannes Gutenberg-Universität, 55099 Mainz, Germany. [9] Present address: The Institute of Photonic Sciences, 08860 Castelldefels, Spain. [10] Present address: Institute for Bioengineering of Catalonia, The Barcelona Institute of Science and Technology, 08028 Barcelona, Spain. Correspondence and requests for materials should be addressed to D.A.B. (email: barskiy@berkeley.edu)

Chemical exchange phenomena are fundamental in aqueous-phase reactions, for example, in biochemistry of living organisms[1]. These reactions are characterized by reaction-exchange rates that can be quantitatively measured with techniques like nuclear magnetic resonance (NMR)[2]. NMR is indeed well placed for studying reaction rates: (i) it allows chemical sensing of a wide range of dynamic timescales, from picoseconds (correlation time of molecular motion) to days (reaction half-lives)[3,4], (ii) it differentiates chemical species through their resonance frequencies (chemical shifts and $J$-couplings)[5,6] and (iii) it probes the chemical composition of studied samples in a nondestructive manner[7].

Despite the tremendous utility of conventional high-field NMR, it has several limitations. First, it requires high magnetic fields (typically several Tesla, often achieved by using super-conducting magnets) and, therefore, is costly and hard to transport. Second, cryogens needed for operation of an NMR magnet necessitate advanced research infrastructure[8]. While NMR can also be performed in transportable permanent magnet systems (1–2 T) with reduced sensitivity and resolution, it is possible to study the behavior of nuclear spins in the complete absence of an applied field. This form of NMR, also known as zero- to ultralow-field (ZULF) NMR, has the ability to provide chemical resolution in mixtures (due to the narrow spectral lines arising from the absence of magnetic-field gradients and long coherence times)[9,10] with a high sensitivity ($10\,\mathrm{fT/Hz^{0.5}}$), without a need for strong, persistent magnetic fields[11,12]. Furthermore, the direct detection of spin−spin coupling spectra (also called $J$-spectra) at zero field provides an opportunity to probe the collective effect of exchange upon two or more nuclear spins, rather than probe a single nucleus as in conventional high-field NMR. Recent advances in atomic magnetometry have facilitated studies of zero-field NMR since highly sensitive magnetic resonance instruments can be built in a way that is inexpensive and highly portable[13–17].

In a more fundamental context, chemical exchange influences the dynamic state of the nuclear spin system, and, therefore, the timescale over which nuclear spins can store information. During chemical exchange, nuclear spins randomly make/break connections with other members of the ensemble and therefore decohere, accelerating the tendency of magnetic spin order towards thermal equilibrium. The nature of these relaxation processes is determined by many factors, including magnitude of the spin quantum number, gyromagnetic ratio, and spin−spin couplings, as well as the symmetry of the spin system[2]. Due to this reason, chemical exchange can have a negative impact in systems where zero-field NMR is seen as advantageous detection modality, and therefore, studying the effect of chemical exchange on ZULF NMR spectra is certainly warranted.

In this work, we investigate the effect of chemical exchange on ZULF NMR spectra by studying aqueous solutions of [$^{15}$N] ammonium ($^{15}\mathrm{NH_4^+}$) under varying pH conditions. Proton ($^1$H) dissociation rates in $^{15}\mathrm{NH_4^+}$ span several orders of magnitude (from ~$2\,\mathrm{s^{-1}}$ to more than $40{,}000\,\mathrm{s^{-1}}$) as pH increases from −1 to 5. For pH levels above 1, ZULF NMR peaks of $^{15}\mathrm{NH_4^+}$ vanish. We explain the observed behavior using a theoretical model that considers the interplay between chemical exchange and nuclear spin dynamics. We additionally rationalize these findings by the Markov-chain analysis of the "nuclear-spin memory" in the system and demonstrate that only ~10 proton dissociation −association events in $^{15}\mathrm{NH_4^+}$ are enough to lose nuclear-spin correlations necessary for observation of the ZULF NMR spectrum. Based on our observations, we formulate general advantages and limitations of ZULF NMR for the analysis of exchanging chemical species. To demonstrate feasibility of studying relevant biochemical substrates at zero magnetic field, we perform ZULF NMR measurements of [2-$^{13}$C]pyruvic acid

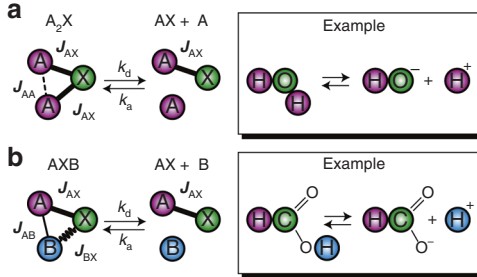

**Fig. 1** Chemical exchange scenarios important in the context of zero- to ultralow-field (ZULF) NMR. **a** Exchange affecting the entire $J$-coupled network. All atoms of the molecule can break chemical bonds between each other. Practically relevant examples include exchange in symmetric molecules such as $H_2O$, $NH_4$, etc. **b** Exchange affecting a subsystem of $J$-coupled network. One part of the spin system exchanges while the rest of the molecule stays intact. A practically relevant example is proton exchange in molecules containing more than a single coupled nucleus. Note that once dissociated, hydrogen (light blue) can attach to a different molecule, i.e., the exchange is intermolecular

hyperpolarized via dissolution dynamic nuclear polarization (dDNP). Hyperpolarized pyruvate is used as a molecular probe to study biochemical transformations in vivo using magnetic resonance imaging (MRI) and spectroscopy[18]. Given these findings, we suggest ways to detect hyperpolarized molecules by portable ZULF NMR spectrometers in places where conventional, high-field NMR is hard to implement.

## Results

**Chemical exchange regimes**. We distinguish two chemical exchange scenarios important for further discussion.

The first scenario is exchange affecting the entire $J$-coupled network (Fig. 1a); molecules can exchange any of their chemically equivalent nuclear spins with other molecules and/or solvent. The simplest example using Pople nomenclature for coupled spin systems would be the exchange $A_2X \rightleftarrows A + AX$ (here A and X denote different types of nuclear spins, for example $^1$H and $^{13}$C). This scenario applies to proton exchange in ammonium, water and other symmetric molecules where a labile spin group (e.g., $H^+$) is present.

The second scenario is exchange affecting a subsystem of $J$-coupled network (Fig. 1b), where one part of the molecule exchanges while the rest of the molecule retains its structure. This corresponds to the situation when an AXB spin system dissociates and forms two smaller systems AX and B. An example is formic acid with AX being the $^1$H$^{13}$C fragment and B being the exchangeable acidic proton. In more complex spin systems, such as [2-$^{13}$C]pyruvic acid, chemical exchange can occur simultaneously at more than one distinct (chemically inequivalent) groups in the molecule, see below.

The effect of chemical exchange on the coherent spin dynamics (e.g., apparent resonance frequencies of the lines in NMR spectra) depends on the rate of exchange relative to the difference in NMR frequencies of exchanging sites. Therefore, it is clear that ZULF NMR spectra depend significantly on the type of the exchange scenario taking place. However, this effect will manifest itself in ZULF NMR in a different way compared to high-field NMR. For example, in both of the scenarios above, a high-field NMR spectrum of the species X will always give an observable signal and only the $J$-splitting pattern will change depending on the exchange scenario and specific exchange rate. This is because high-field NMR detects precession of a single nuclear species, i.e., $^1$H, $^{13}$C, $^{15}$N. On the other hand, the signal in ZULF NMR is a

result of a time-dependent coherence involving different types of nuclear species. Dissociation of any part of the molecule during the acquisition time will significantly affect the whole spectrum.

We note that other exchange scenarios are possible as well. For example, intramolecular rearrangements (such as conformational isomerism in hydroxylamines)[19] can alter $J$-couplings in the system without breaking chemical bonds. Such systems have not been studied in this paper but may be described using a similar theoretical model, see below.

**Calculation approach**. In order to describe the effect of chemical exchange on zero-field NMR spectra quantitatively, we developed a model that combines chemical exchange kinetics and nuclear spin dynamics[20–25]. In this model, we assume the following exchange scheme:

$$A + B \rightleftharpoons C. \qquad (1)$$

Here, two chemical species, A and B, can reversibly combine to form a species C. Differential equations describing the chemical kinetics of this system can be written in the matrix form as:

$$\frac{d}{dt}\begin{pmatrix} [A] \\ [C] \end{pmatrix} = \begin{pmatrix} -k_a[B] & +k_d \\ +k_a[B] & -k_d \end{pmatrix}\begin{pmatrix} [A] \\ [C] \end{pmatrix}. \qquad (2)$$

Here [A], [B], and [C] are concentrations of A, B, and C; $k_d$ and $k_a$ are dissociation and association reaction-rate constants, respectively. Description (2) is convenient when one of the reagents (e.g., B) is present in excess. In this case one can define $W_a = k_a[B]$ as a rate of A's association and this quantity can be considered constant during the course of the reaction. In equilibrium, concentrations of species in the system are determined by the equilibrium constant $K = k_d/k_a$ (such definition of the equilibrium constant is convenient for the reasons explained below).

A similar approach can be applied for simulating nuclear spin dynamics in the presence of chemical exchange. The master equation describing evolution of the total nuclear spin system is:

$$\frac{d}{dt}\begin{pmatrix} \hat{\rho}_A \\ \hat{\rho}_C \end{pmatrix} = \begin{pmatrix} \widehat{\widehat{L}}_A - W_a\widehat{\widehat{1}}_A & +k_d\widehat{\widehat{T}}_B^{(C)} \\ +W_a\widehat{\widehat{D}}_B^{(A)} & \widehat{\widehat{L}}_C - k_d\widehat{\widehat{1}}_C \end{pmatrix}\begin{pmatrix} \hat{\rho}_A \\ \hat{\rho}_C \end{pmatrix}, \qquad (3)$$

where $\hat{\rho}_A$ and $\hat{\rho}_C$ are the density matrices corresponding to the species A and C, respectively. Liouvillian superoperators $\widehat{\widehat{L}}_A = -i\widehat{\widehat{H}}_A + \widehat{\widehat{R}}_A$ and $\widehat{\widehat{L}}_C = -i\widehat{\widehat{H}}_C + \widehat{\widehat{R}}_C$ describe coherent evolution (defined by Hamiltonian superoperators[26], $\widehat{\widehat{H}}_i = \left[\widehat{H}_i, \cdot\right]$) and incoherent relaxation (defined by relaxation superoperators, $\widehat{\widehat{R}}$) of nuclear spins in molecules A and C. Matrix $\widehat{\widehat{T}}_B^{(C)}$ is a partial trace operator acting on the density matrix $\hat{\rho}_C$ and resulting in the removal of subsystem B. The matrix $\widehat{\widehat{D}}_B^{(A)}$ is a direct-product superoperator representing formation of C from A as a result of adding subsystem B to A. Note that matrices $\widehat{\widehat{T}}_B^{(C)}$ and $\widehat{\widehat{D}}_B^{(A)}$ are not square and depend on the dimensions of the corresponding subspaces[26]. In our approach, $\hat{\rho}_A$ and $\hat{\rho}_C$ are columns produced from corresponding square matrices by column-wise concatenation and the spin state of B ($\hat{\rho}_B$) depends on the experiment and its value is kept constant. For example, an unpolarized ensemble at zero magnetic field can be treated as $\hat{\rho}_B = \widehat{\widehat{1}}_B/\text{Tr}\left\{\widehat{\widehat{1}}_B\right\}$ (see below). We also note that here we do not include terms responsible for equilibrium thermal polarization since they do not affect spin evolution and are equal to zero in the absence of the magnetic fields. Equation (3) is analogous to Bloch−McConnell equations used for calculating the effect of chemical exchange in high-field NMR[27]. However, while the Bloch

−McConnell equations describe the evolution of nuclear magnetization for systems with uncoupled spins, a treatment of coupled multispin systems is necessary here since spin−spin couplings are the key for observing ZULF NMR $J$-spectra. Equations (3) can be solved numerically, and a straightforward approach can be applied to systems in chemical equilibrium. In this case, the combined matrix $\widehat{\widehat{M}}$ is time-independent and the solution for the density matrix $\hat{\rho} = \begin{pmatrix} \hat{\rho}_A \\ \hat{\rho}_C \end{pmatrix}$ at time $t$ is given by exponentiation:

$$\frac{d}{dt}\hat{\rho} = \widehat{\widehat{M}}\hat{\rho} \Rightarrow \hat{\rho}(t) = e^{\widehat{\widehat{M}}t}\hat{\rho}(0). \qquad (4)$$

As an example, let us consider proton exchange in aqueous acidic solutions. In these systems, protons can jump between different molecules present in solution including the solvent. Using our chemical assignments, acid concentration is referred to as [C], $[H^+] = [B]$, and concentration of the conjugate base is [A]. The equilibrium constant for proton dissociation is known as $K_a$ and it determines the molar fractions of the corresponding forms: $x_C = 10^{-pH}/(10^{-pH} + K_a)$ is the molar fraction of the acidic form (here pH $= -\log_{10}([B])$), and $x_A = 1 - x_C = K_a/(1 + K_a)$ is the molar fraction of its conjugate base form. Therefore, we choose normalization such that $\text{Tr}\{\hat{\rho}_A\} = x_A$ and $\text{Tr}\{\hat{\rho}_C\} = x_C$. The convenience of the set of Eq. (3) lies in the fact that the state of freely exchanging protons in solution can be considered uncorrelated and unpolarized, i.e., $\hat{\rho}_B = \widehat{\widehat{1}}_B/\text{Tr}\left\{\widehat{\widehat{1}}_B\right\}$ (here $\widehat{\widehat{1}}_B$ is the unit matrix) and, therefore, embedded into the matrix $\widehat{\widehat{M}}$. This is reasonable since $T_1/T_2$ relaxation of exchanging protons $[H^+]$ in aqueous solutions is known to be fast on the relevant timescales (see below)[28]. Therefore, a combined matrix $\widehat{\widehat{M}}$ describing chemical exchange and spin dynamics is time-independent. According to Eq. (4), this allows propagating the system given known initial conditions by exponentiation.

Relaxation effects are incorporated in the simulation by including Redfield relaxation superoperators into the matrix $\widehat{\widehat{M}}$. A model of the local fluctuating fields was used[29–31], in which the relaxation superoperator is derived from the known values of $T_1$ at high magnetic field (no additional data on the molecular motion are necessary)[32]. The effect of radiofrequency (RF) pulses and other magnetic field manipulations can be also considered. Simulations were performed using the MOIN spin library (available online)[33] that has been shown to efficiently model nuclear spin dynamics in chemically exchanging systems[34]. Details of calculating NMR spectra and relaxation of polarization will be described elsewhere.

**Effect of chemical exchange on ZULF NMR $J$-spectra**. We performed numerical simulations with the goal of analyzing the effect on ZULF NMR spectra of two different exchange scenarios discussed above (Fig. 2). The first case is an AX spin system that can dissociate into parts, A and X. For simplicity, dissociation and association rate constants were considered to be equal ($k_d = k_a[X] = k$). One can see that an increase in the dissociation rate results in broadening of the NMR line at $J_{AX}$ in the zero-field spectrum until it becomes too broad to be observable (Fig. 2a). The second case corresponds to an AXB spin system with the spin B undergoing chemical exchange. In this case, initial splitting determined by the $J$-coupling topology of an AXB system gradually disappears as the exchange rate increases (Fig. 2b). Remarkably, all NMR lines collapse into a single line (corresponding to AX spin system) when the exchange rate is fast. Since

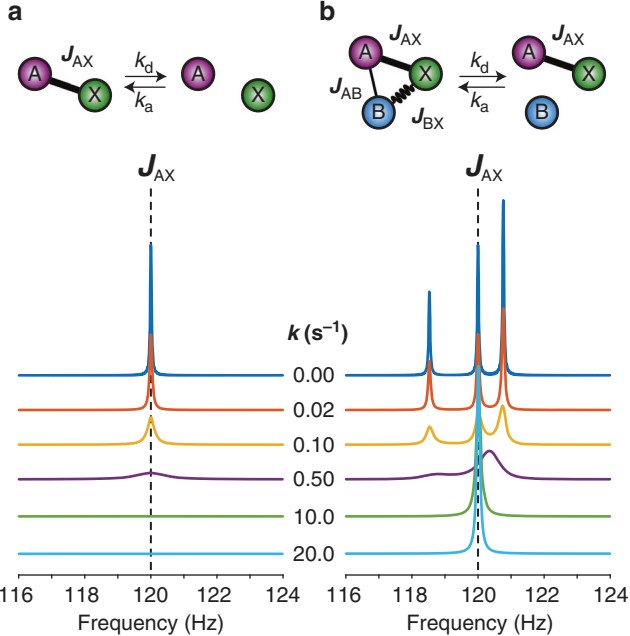

**Fig. 2** Simulated ZULF NMR spectra as a function of chemical exchange rate. **a** Exchange in the AX spin system (A is ¹H spin, X is ¹³C spin, $J$-coupling is 120 Hz). One can clearly see signal broadening as a function of dissociation exchange rate. **b** Exchange in the AXB spin system (X is ¹³C spin, A and B are ¹H spins, $J_{AX} = 120$ Hz, $J_{AB} = 3$ Hz, $J_{XB} = 0$ Hz). In both simulations, the forward reaction rate and the reverse reaction rate were considered to be equal: $k = k_d = k_a[X]$ and $k = k_d = k_a[B]$ for the case (**a**) and (**b**), respectively

initially we assumed equimolar concentrations of AXB and AX (since both dissociation and association rate constants were set equal in the simulation, resulting in $x_{AX} = x_{AXB} = 0.5$), half of the line's intensity belongs to AX and another half—to AXB.

We note that this case represents a common scenario, the proton exchange in solutions of organic molecules ¹³C-labeled in carboxylic position, e.g., [¹³C]-formic acid. A proton (B) originating from the carboxylic group is undergoing frequent dissociation and association events with the corresponding conjugate base and/or water and, therefore, its effect on spin dynamics of the rest of the spins (AX) becomes negligible when dissociation rate is higher than $J_{AB}$ and $J_{XB}$ spin−spin couplings. The fragment AX (¹H¹³C) is stable and not affected by proton exchange resulting in the NMR line at 222 Hz[17]. Other organic acids, alcohols, and ketones can also be subject to this exchange mechanism (intermolecular exchange involving chemically inequivalent nuclear sites).

Despite the fact that chemical exchange affects high-field NMR and ZULF NMR spectra differently, some analogies can still be found. Consider, for example, ¹H NMR spectrum of ethanol. In anhydrous ethanol, the rate of −OH proton exchange is small (<10 Hz), and, therefore all three peaks (−OH, −CH₂-, and −CH₃) are observable in ¹H NMR[2]. Proton exchange accelerated by addition of water results in a significant broadening of −OH resonance (similar to the signal degradation in Fig. 2a) and disappearance of the multiplet structure corresponding to the $J$-coupling with this proton for −CH₂- group (similar to Fig. 2b). This demonstrates that, despite apparent complexity, physical intuition for the analysis of ZULF NMR spectra of chemically exchanging systems can be gained from conventional high-field NMR textbook examples.

**pH-dependent chemical exchange in ¹⁵NH₄⁺.** To study the effect of chemical exchange rate on ZULF NMR spectra we chose ¹⁵N-labeled ammonium (¹⁵NH₄⁺), whose spin system undergoes exchange affecting the entire $J$-coupled network (Fig. 3a). Since the molecule contains four chemically (and magnetically) equivalent hydrogens, each of the four ¹H atoms have equal dissociation probability. This chemical system, in a sense, is unique because it allows one to control hydrogen exchange rates simply by varying pH while keeping the same concentration of ¹⁵NH₄⁺ in solution. Indeed, when pH is less than 7, [¹⁵NH₄⁺] is the dominant form in solution (Fig. 3b). At the same time, the dissociation rate constant $k_d$ depends dramatically on pH and spans several orders of magnitude, from approximately 2 s⁻¹ to more than 40,000 s⁻¹ (Fig. 3c).

When detected at high-field (18.8 T) and in a highly acidic environment (pH < 0), the ¹⁵N NMR spectra of ammonium-¹⁵N (6 M solution in H₂O) show a 1:4:6:4:1 quintet with characteristic splitting of 73.4 Hz originating from heteronuclear $J$-coupling between ¹⁵N and each of four hydrogen nuclei. Upon increasing pH level, all five resonances broaden and eventually merge into a single narrow peak clearly indicating that the increase of pH-dependent proton dissociation rates bring the system into a motional-narrowing regime (Fig. 3c). While these changes of NMR multiplets are expected, it is a rare case of a $J$-resolved NMR multiplet recorded under precisely controlled exchange-rate conditions (by setting pH value) and being collected for a sample with constant concentration of the studied chemical (¹⁵NH₄⁺).

To quantify the observed behavior, we performed simulations of the high-field ¹⁵N NMR spectra using the approach described above (see Calculation approach). Since NMR parameters of the molecule are known, only the kinetic rate constant $k_d$ was varied in the simulation ($k_a$ and, hence, $W_a$ were determined from $k_d$ and pH, see SI). When $k_d$ is much smaller than $J_{NH}$, increase in $k_d$ results in broadening of all of the peaks in the multiplet as characterized by the increase of their full width at half height (FWHH). Further increase in $k_d$ results in merging of all peaks into a single line that becomes narrower after the coalescence point at pH ~3.

We compared two algorithms of calculating the effect of proton dissociation on high-field ¹⁵N NMR spectra by analyzing FWHHs of the simulated peaks (Supplementary Figs. 3–5). The first algorithm assumed the case where all four protons dissociate from the ¹⁵N atom at the same time. While obviously simplistic, this approach reproduces the general character of the observed phenomenon, i.e., broadening and merging of the NMR lines into one peak upon further increasing $k_d$. The second algorithm considered a more realistic scenario, i.e., exchange of only one (random) proton at a time; the result of simulation is shown in Fig. 3c. Surprisingly, at low exchange rates ($k_d \ll J_{NH}$), FWHH of the peaks is similar for both calculation approaches and is predominantly determined by the intrinsic NMR linewidth (Supplementary Tables 2 and 3); however, for fast exchange rates ($k_d \gg J_{NH}$), the same value of FWHH is obtained when $k_d$ for the first simulation algorithm (simultaneous exchange of all protons) is set four times smaller than $k_d$ in the second simulation approach (exchange of a random proton).

Developing an analytical model for the effect of chemical exchange on $J$-coupled spectra lies outside of the scope of this paper. However, our simulation approach can be used to gain important information about details of the exchange mechanism. For example, the calculated amplitude of the middle line in high-field ¹⁵N NMR spectra of ¹⁵NH₄⁺ is lower for the case when a random proton dissociates (compared to the calculation considering simultaneous dissociation of all four protons), and fits

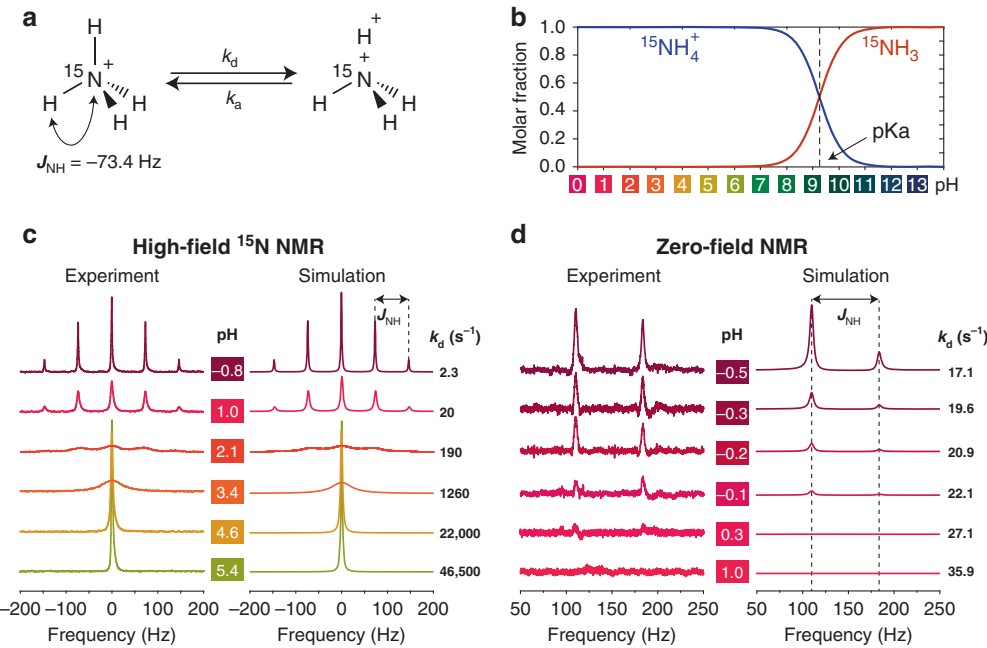

**Fig. 3** Chemical exchange in the $^{15}$N-labeled ammonium. **a** Molecular diagram of the chemical exchange in $^{15}NH_4^+$. Each of the four hydrogen atoms can dissociate with a kinetic rate constant $k_d$ and associate back with a kinetic rate constant $k_a$ (defined through the equilibrium constant $K_a = k_d/k_a$; $pK_a = -\log_{10}(K_a)$). **b** Molar fractions of different forms of ammonia in solution as a function of pH. Color code is based on universal pH indicator. Note that for pH < $pK_a$ ammonium is predominantly present in the form of $^{15}NH_4^+$ (rather than $^{15}NH_3$). **c** High-field (18.8 T) $^{15}$N NMR spectra as a function of pH. **d** ZULF NMR spectra of $^{15}NH_4^+$ as a function of pH. Note the difference in the pH range for the high-field and ZULF spectra

the experimental data better (Supplementary Fig. 3). While the fact that only one proton dissociates at a time may have seemed intuitive for $^{15}NH_4^+$, it might not always be clear which exchange scenario is operative for more complex chemically exchanging systems. Comparing simulated NMR spectra under different exchange scenarios with those from experiment may, therefore, help to elucidate the exact exchange mechanism for a system under study.

We now turn to the ZULF NMR spectra of ammonium solutions. Spectra recorded for very acidic (pH < 0) solutions show two distinct peaks at $3J_{NH}/2 = 110.4$ Hz and $5J_{NH}/2 = 183.6$ Hz as expected from the energy-level analysis of $A_4X$ spin systems (one peak corresponds to the transitions in the manifold with total hydrogen spin 1, and the other one with total hydrogen spin 2)[13]. However, instead of gradual broadening of NMR peaks on the pH scale comparable to the high-field spectra, abrupt decay in the signal amplitude is observed for zero-field spectra with pH levels increasing from −0.5 to 1.0 (Fig. 3d). For pH values above 1, no zero-field NMR signal of $^{15}NH_4^+$ was detected. Our initial attempts to simulate such an abrupt signal vanishing were not successful no matter what exchange mechanism was considered. However, it was found that the key to understanding the reasons of such signal disappearance lies not in the exchange mechanism but rather in the detection protocol. Indeed, initial nuclear spin polarization is necessary before the sample's signal can be detected in ZULF NMR. In our experiments, ammonium samples were first prepolarized with a neodymium magnet (~2 T) and then pneumatically shuttled to the zero-field region. It turns out that the time spent by the sample between the high-field polarization and zero-field detection (~0.5 s, see Methods) leads to equilibration of the populations in $^{15}NH_4^+$, given the fact that free protons in solution (or, more precisely, hydrated forms such as Eigen and Zundel structures, etc.)[35,36] can be considered unpolarized in low magnetic fields. Therefore, in a nutshell, it is

the number of dissociation−association events during the shuttling time that affects the nuclear spin memory, and therefore, the signal that can be detected by ZULF NMR spectrometer.

To obtain an intuitive understanding of such signal decay, we performed a Markov-chain analysis of the effect of proton exchange on polarization of spins in $^{15}NH_4^+$ (Supplementary Figs. 6–8). We find that only ten exchange events are necessary for $^{15}NH_4^+$ to lose ~99.6% memory of its initial state (i.e., if the protons in $^{15}NH_4^+$ are initially prepared in the $|\alpha\alpha\alpha\alpha\rangle$ state) as indicated by a significant increase in calculated spin entropy (Supplementary Fig. 8). More complex spin states including entanglement of all or part of the spins in the system would result in even faster memory loss. This allows to explain why shuttling leads to a decrease in the $^{15}NH_4^+$ signal amplitude in ZULF NMR spectra. Molecular lifetime of the ammonium (i.e., time the molecule spends between subsequent proton dissociation−association events) is determined by $1/k_d$. From the data shown in Fig. 3c, for pH = −0.8, such lifetime is ~0.5 s, and it decreases to ~20 μs for pH = 5. This means that, on average, if more than ten exchange events happen during the time spent by the ammonium sample in the zero-field chamber (either at zero-field or at low magnetic field of the guiding solenoid), ZULF NMR signal would vanish. Indeed, $k_d$ on the order of 20 s$^{-1}$ is enough to destroy the signal if the shuttling time is ~0.5 s. While the lifetime of $^{15}$N spin polarization ($T_1$) is longer than the molecular lifetime of $^{15}NH_4^+$, it is predominantly $^1$H polarization that determines ZULF NMR signal—due to much higher gyromagnetic ratio of protons compared to $^{15}$N nuclei. We also note that precise temperature control of the samples in ZULF NMR spectrometers used in this work was not possible. Therefore, we expect temperature of the sample to be ~40 °C (compared to 25 °C for high-field NMR measurements) as estimated based on separate measurements[37,38], which additionally increases dissociation

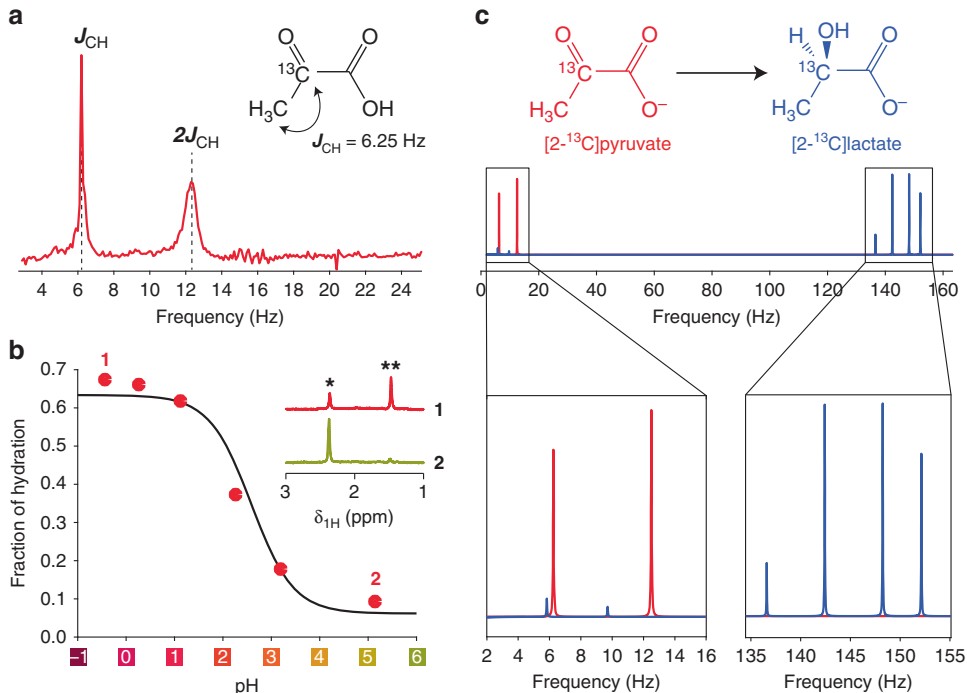

**Fig. 4** Chemical exchange in the [2-$^{13}$C]pyruvic acid. **a** ZULF NMR spectrum of [2-$^{13}$C]pyruvic acid (single scan) hyperpolarized via dissolution dynamic nuclear polarization (dDNP). **b** Red circles—fraction of hydration for the pyruvic acid and its conjugate base measured with benchtop $^{1}$H NMR (60 MHz) as a function of pH. Black line is fit with Supplementary Eq. 18. The inset demonstrates spectra at pH = −0.4 and 5.2, single and double asterisks denote $^{1}$H NMR resonances of CH$_3$ group of the nonhydrated and hydrated forms of pyruvic acid, respectively. **c** Simulated ZULF NMR spectrum of [2-$^{13}$C]pyruvate and [2-$^{13}$C]lactate. Biochemical conversion of pyruvate to lactate is an important target for studies of metabolism in vivo. ZULF NMR detection of hyperpolarized molecules may endow narrow (sub-Hz) resonances separated by more than 100 Hz, a typical carbon-proton $J$-coupling value. Note that in the simulation, we did not consider relaxation/decoherence effects

rates. Taken together, this explains significantly faster signal disappearance with increasing pH for ZULF NMR spectra as compared to conventional high-field NMR.

**ZULF NMR detection of hyperpolarized [2-$^{13}$C]pyruvic acid.** In order to demonstrate the potential of ZULF NMR for studying chemically exchanging systems relevant for biomedical applications, we performed measurements of hyperpolarized [2-$^{13}$C] pyruvic acid at zero magnetic field. Hyperpolarization by dDNP (see Methods) superseded time-consuming prepolarization and signal averaging, thus accelerating ZULF NMR signal acquisition by a factor of ~10,000 (hundreds of scans would have been needed to detect a signal with the same SNR, Supplementary Fig. 1). Furthermore, this is the first demonstration of compatibility of ZULF NMR with hyperpolarization by dDNP.

Figure 4a shows a zero-field NMR spectrum of dilute (80 mM) [2-$^{13}$C]pyruvic acid collected in a single scan after the transfer of hyperpolarized [2-$^{13}$C]pyruvic acid from dDNP polarizer to the portable ZULF NMR device. Two peaks, one at $J_{CH}$ and one at $2J_{CH}$, are clearly observable as expected for the A$_3$X nuclear spin system corresponding to the transitions in the manifold with the total proton spin 1/2 and total proton spin 3/2, respectively ($J_{CH} \approx 6.25$ Hz is a two-bond spin−spin coupling constant between $^{13}$C and $^{1}$H nuclei). The fact that no other peaks are observable in the spectrum indicates the stability of the $^{13}$C−CH$_3$ fragment and shows that chemical exchange events (e.g., hydration and proton exchange) are fast on a timescale of $1/J_{CH}$ (see the case discussed in Fig. 2b).

The sensitivity of the prototype portable spectrometer was estimated to be in the range of 500−1000 fT/Hz$^{1/2}$ which is two orders of magnitude worse than the recently achieved sensitivity of 10−20 fT/Hz$^{1/2}$ on similar devices[12]. Nevertheless, even given

significant losses of sample polarization due to nonoptimal transfer time and low detection sensitivity of the ZULF spectrometer, the collected spectrum has a high SNR of ~30 and ~10 for the peaks at $J_{CH}$ and $2J_{CH}$, respectively. Significant improvement in detection sensitivity is expected for an optimized ZULF NMR device with integrated fluid transfer path, which would minimize relaxation losses during the transfer.

Hyperpolarized pyruvate is widely used by the biomedical NMR/MRI community as a marker of metabolic changes associated with a number of diseases and response to treatment. Enzymatic conversion of pyruvate with a $^{13}$C-label placed in the carboxyl position ([1-$^{13}$C]pyruvate) into lactate and alanine, for example, has been used to differentiate between healthy and diseased tissue in cancer[39–41] or diabetes[42,43]. By placing the $^{13}$C-label in the ketone position ([2-$^{13}$C]pyruvate), the chemistry of oxidative phosphorylation can be tracked and spectral information on metabolites of the Krebs cycle, ketone bodies or lipogenesis can be obtained[44]. [2-$^{13}$C]pyruvate is particularly relevant in the study of cardiac metabolism[45,46]. In our work, we used [2-$^{13}$C]pyruvic acid, obtaining a $J$-spectrum in a reasonable frequency range of 5−15 Hz which is free from background noise and/or other peaks. The relevant $J$-coupling of 1.3 Hz in [1-$^{13}$C] pyruvic acid is too small to be efficiently detected in ZULF with the detection scheme employed in our work; its NMR signal would overlap with the zero-frequency peak originating from the static magnetization of the sample.

The two zero-field NMR peaks at $J_{CH}$ and $2J_{CH}$ observed for [2-$^{13}$C]pyruvic acid are similar in their integral values (28 a.u. vs. 26 a.u.) but have quite different FWHHs (0.14 Hz vs. 0.71 Hz) resulting in ~2.7 times difference in their amplitude (Fig. 4a). A broader line is a result of faster decay of spin coherence precessing at the frequency $2J_{CH}$ compared to spin coherence

precessing at the frequency $J_{CH}$ at zero magnetic field. The exact reason of this phenomenon is not completely clear and we have three possible explanations (see below).

The first possible explanation of the broader line at $2J_{CH}$ involves chemical exchange of pyruvic acid with water (hydration). It is well known that both pyruvic acid and pyruvate anion can undergo reversible attachment of water resulting in 2,2-dihydroxypropionic acid (and 2,2-dihydroxypropionate anion) and this process is pH dependent[47]. We ran additional benchtop $^1H$ NMR (60 MHz) measurements to identify the hydration fraction of pyruvic acid at different pH values by analyzing the values of $^1H$ NMR integrals corresponding to $CH_3$ groups in hydrated and nonhydrated forms (Fig. 4b). Our results reproduce observations from literature indicating that ~40% of pyruvic acid (and/or pyruvate) are present in the hydrated form at pH = 2.5[47,48]. This pH is an estimated value for our sample at the moment of its zero-field NMR detection (see Methods). However, carrying out numerical simulations of ZULF NMR spectra for these molar fractions of hydrated and nonhydrated forms and assuming chemical exchange as a $(A_3X)B_2 \rightleftharpoons A_3X + 2B$ process (here $(A_3X)B_2$ is 2,2-dihydroxypropionic acid, $A_3X$ is pyruvic acid and $2B$ are protons of water) did not give different FWHHs for the resulting peaks upon chemical-exchange-induced broadening (Supplementary Figs. 9 and 10). Therefore, we assume that chemical exchange alone cannot explain the observed differential peak broadening.

The second possible explanation considers the presence of magnetic-field gradients across the sample (i.e., field inhomogeneity). It is indeed the case that magnetic field applied along any direction would lead to a larger splitting of the peak at $2J_{CH}$ compared to the peak at $J_{CH}$. Supplementary Fig. 11 shows that a magnetic field applied along $z$-direction (parallel to the magnetometer sensitive axis) leads to the splitting of the line at $2J_{CH}$ while the line at $J_{CH}$ is intact. Nonuniform distribution of the field (i.e., a field gradient) would lead to overlapping different multiplets resulting in overall wider line. A static field perpendicular to the sensitive axis splits both peaks; however, splitting of the peak at $2J_{CH}$ is still larger than splitting at $J_{CH}$[11]. In general, any magnetic interaction of the molecule will affect the line at $2J_{CH}$ more significantly than the line at $J_{CH}$ due to larger effective magnetic moment for the transitions involving the manifold with total hydrogen spin 3/2. Ongoing chemical exchange (hydration) and field gradients across the sample both lead to smoothing of the spectral features and may explain differential line broadening. We note that the presence of paramagnetic impurities (e.g., not completely filtered OX063 radicals used for dDNP process) can also contribute to the observed phenomenon.

The third possible explanation is faster intrinsic relaxation of the corresponding transitions at zero field. We compared two ways to simulate nuclear spin relaxation by embedding Redfield superoperators in Eq. 3. Our results show that depending on the simulation approach, either first or the second peak can be broader. Indeed, using the relaxation model of fluctuating magnetic fields[32,33] gives larger broadening to the peak at $J_{CH}$ while the model of dipole−dipole relaxation makes the peak at $2J_{CH}$ to appear broader (Supplementary Fig. 12).

It is likely that a combination of some or all of the above mechanisms can be responsible for the observed behavior and further studies are warranted to elucidate the exact nature of this effect.

Our study demonstrates that ZULF NMR spectra provide unique information about relaxation and/or chemical exchange phenomena observable directly in 1D spectra. Other techniques, including 2D ZULF NMR methods, decoupling and selective pulses can offer additional spectroscopic capabilities complementing 1D data.

**Hyperpolarized ZULF NMR of chemically exchanging systems**. Findings of this study are important both in the context of ZULF NMR detection as well as for understanding the dynamics of hyperpolarized NMR probes. The fact that the species undergoing exchange affecting the entire $J$-coupled network (Fig. 2a) can be completely unobservable in ZULF NMR has important implications. It means that only signals of chemicals undergoing exchange in the regime $k_d < |J_{NH}|$ can be detected in ZULF NMR, which can be seen as a significant disadvantage of this detection modality. However, the same fact can be advantageous for eliminating the signal from solvents such as water or when the studied system is complex (e.g., cell cultures or bioreactors) and observation of only selected chemical pathways is desired. Moreover, if the chemical exchange rate matches relevant $J$-couplings in the system, it can lead to significantly increased polarization decay in the strong-coupling regime (when the $J$-coupling equals or exceeds to the difference of the Zeeman energies for the coupled spins). This is important for applications of dDNP in situations when initially highly polarized samples are transferred from the site of their production through low-magnetic-field regions (often Earth's magnetic field) for subsequent in vivo injection. Recent work[49] demonstrated that deuteration of exchangeable protons in [5-$^{13}$C]glutamine, [6-$^{13}$C] arginine, and [$^{13}$C, $^{15}$N$_2$]urea significantly increases the lifetime of the $^{13}$C polarization. Our study shows that not only reduced contribution of dipole−dipole relaxation but also changes in chemical exchange rates and $J$-couplings can contribute to the increase (or decrease) of polarization lifetime during the time hyperpolarized samples spend in the ambient laboratory magnetic field.

Our simulations and experimental results are also relevant in the context of signal amplification by reversible exchange (SABRE) hyperpolarization technique, especially its near-zero-field variant[50,51]. In this technique, the source of nuclear spin order, parahydrogen (*para*-H$_2$), is brought in contact with a to-be-polarized target and polarization transfer is enabled due to the chemical exchange between the source and the target on an iridium-based metal complex. Our results may explain why the signal of the polarization transfer complex was never observed in ZULF NMR; indeed, the exchange rates are too fast for the relevant $J$-couplings to be manifested at zero field[52]. This is not necessarily a disadvantage because it allows to clear out the ZULF spectra from the signatures of reaction intermediates and focus on the dynamic of the substrate alone.

To demonstrate the potential of ZULF NMR in the study of biologically relevant processes, we simulated the ZULF NMR spectrum of [2-$^{13}$C]lactate using $J$-coupling values found in the literature[53]. The main features of the ZULF NMR spectrum of lactate lie in the high-frequency region (130−150 Hz) as determined by the spin-coupling topology of lactate (the main contributing factor is the direct $^{13}$C-$^1$H coupling) as opposed to the spectrum of [2-$^{13}$C]pyruvate with the spectral features located in the low-frequency region (Fig. 4c). This spectroscopic differentiation of two seemingly similar chemicals is an important advantage of the ZULF compared to high-field NMR. Indeed, high-field analysis of hyperpolarized probes is typically based on the chemical shift difference between the molecules undergoing transformation which is significantly affected by magnetic field inhomogeneity of the sample. We note that not the $J$-coupling values themselves but rather $J$-coupling topology of the molecules determines the differences in their ZULF NMR patterns.

Overall, ZULF NMR is well suited for studying chemical exchange phenomena as demonstrated in the present study. Practically relevant applications of ZULF NMR if combined with any of the available hyperpolarization techniques may include detection of microscopic biological samples such as cell cultures[54] and reactors with complicated geometry (which are hard to study by conventional NMR due to their size). The recent advent of radicals that can be generated or annihilated on demand can also enhance the synergy between dDNP and ZULF NMR by removing the necessity of the radical filtration step[55–57]. Importantly, the zero-field detection modality is not limited to spectroscopy: an array of magnetometers can in principle be used to generate an image from a series of separate spectroscopic measurements. In this arrangement, an externally hyperpolarized probe can be intravenously injected into a patient and NMR signals can be detected via an array of magnetometers in a manner similar to one recently demonstrated for brain magnetoencephalography[58]. In addition to atomic magnetometers, sensors based to the nitrogen-vacancy (NV) color centers in diamond can be used, whose operation at zero field (relevant for ZULF NMR) was recently demonstrated[59].

## Methods

**High-field NMR of thermally polarized samples.** A stock solution of $^{15}NH_4Cl$ was prepared in a 6 M concentration by dissolving $^{15}NH_4Cl$ (Sigma-Aldrich 299251) in distilled water. Sodium hydroxide (Fisher Chemical, USA), hydrochloric acid (Macron Fine Chemicals, Avantor, USA), or acetic acid (Macron Fine Chemicals, Avantor, USA) at different molarities were used to adjust pH. Sample pH values were measured at room temperature using benchtop pH/ORP meter (HI 2211, Hanna Instruments, Rhode Island, USA) with micro pH combination electrode, glass body (Sigma-Aldrich, St. Louis, MO). Each sample (150 μL for ZULF NMR measurements and 600 μL for high-field NMR measurements) was pipetted into a standard 5 mm NMR tube and flame-sealed before the measurement. High-field $^{15}N$ NMR measurements were performed at room temperature using an 800 MHz Bruker NMR spectrometer.

A stock solution of pyruvic acid (0.2 M, Sigma-Aldrich, 107360) was prepared in pure $D_2O$ (99.9% $^2H$). The pH values were adjusted using sodium hydroxide or hydrochloric acid (Supplementary Table 1). Each sample of pyruvic acid (600 μL) was pipetted into a standard 5 mm NMR tube for $^1H$ NMR measurements performed at room temperature using a 60 MHz Benchtop NMR (NMReady-60PRO, Calgary, Alberta, Canada).

To record the zero-field NMR spectra of the ammonium samples, a home-built ZULF NMR spectrometer ($^{87}Rb$ atomic magnetometer and mu-metal chamber) was used—physical principles of operation, construction, and calibration of the instrument are described in detail in ref. [12]. Each zero-field spectrum is an average of 100 scans, polarization time = 30 s in the field of 2 T, shuttling time = 0.5 s to the zero-field chamber. During the shuttling, a guiding field of 30 μT was applied in the vertical direction with a solenoid wrapped around the shuttling tube. In the zero-field chamber, the field was suddenly turned off (~50 μs) to initiate the evolution of nuclear spin coherence giving rise to the detectable NMR signal.

**ZULF NMR of hyperpolarized samples.** Hyperpolarized [2-$^{13}C$]pyruvic acid was prepared as follows. 15 mM of OX063 radical (GE Healthcare) were added to neat [2-$^{13}C$]pyruvic acid (99% $^{13}C$ enriched, Sigma-Aldrich Isotec, Miamisburg, OH). The liquid was frozen into a glass, then polarized for 45 min at 1.3 K in a commercial dissolution-DNP instrument (3.35 T, Oxford Instruments HyperSense, Abingdon, UK), following a standard protocol (microwave frequency: 94.099 GHz; microwave power: 25 mW; polarized on the positive lobe of EPR line)[60]. Since the narrow-line radical OX063 was used in the sample preparation, we expect mainly $^{13}C$ to be polarized via solid effect. Polarization level of ~15% was estimated from the amplitude of the solid-state $^{13}C$ NMR signal prior to dissolution. The frozen sample was then dissolved in 4 mL of superheated $H_2O$ and the radical removed by filtering (C-18 column, WAT023501). Immediately after sample dissolution and radical filtering, a portion of the final hyperpolarized solution (200 μL, 80 mM [2-$^{13}C$]pyruvic acid, pH < 2.5, 37 °C) was transferred to a 5 mm outer diameter NMR tube in the ambient laboratory magnetic field.

ZULF NMR spectra of hyperpolarized [2-$^{13}C$]pyruvic acid were recorded in a portable version of the zero-field instrument located ~3 m away from the DNP instrument. The sample prepared as described above was inserted by hand into the mu-metal chamber (residual field <1 nT) using a piercing solenoid (100 μT) to maintain orientation of the nuclear magnetization along the vertical axis[12]. The magnetometer signal was recorded (16 s acquisition time, 1000 Hz sampling rate) after nonadiabatically switching off the field of the piercing solenoid (<0.1 ms). Data were Fourier transformed to yield the NMR spectrum. After the first

spectrum was recorded, a fresh NMR tube containing the same volume of hyperpolarized solution was inserted, and the process repeated until there was no signal left (Supplementary Fig. 2). The total time delay between dissolution of the hyperpolarized material and acquisition of the first zero-field NMR spectrum was ~15 s.

## Data availability

All data generated or analyzed during this study are included in this published article (and its supplementary information files).

## Code availability

Simulations were performed using the magnetic resonance open source initiative (MOIN) spin library which is available online[33].

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

## Acknowledgements

We gratefully acknowledge the financial support by NSF CHE-1709944 and funding from the European Union's Horizon 2020 research and innovation program under the Marie Sklodowska-Curie grant agreement number 766402. I.M.-R. has received financial support through the Junior Leader Postdoctoral Fellowship Programme from La Caixa Banking Foundation (project LCF/BQ/PI18/11630020). S.C. and A.A. has received financial support from The Scientific and Technological Research Council of Turkey (TUBITAK). M.R. has received financial support through the ACalNet program. A.N.P. and J.-B.H. acknowledge the financial support by a DFG - RFBR grant (HO 4604/3-1, No. 19-53-12013).

## Author contributions

D.A.B. proposed the study. D.A.B., S.C., A.A., M.R., T.W., and J.B. performed the experiments with ammonia samples. M.C.D.T., I.M.-R., J.K., and D.B.V. conducted zero-field NMR measurements of dDNP-polarized pyruvic acid. D.A.B., A.N.P., and J.-B.H. performed the theoretical simulations. D.A.B. wrote the manuscript with input from all authors. All authors reviewed the manuscript and suggested improvements. D.B. and A.P. supervised the overall research effort.

## Additional information

**Competing interests:** The authors declare no competing interests.

