## [Peer Review File · Nature Communications]

Reviewers' comments:

Reviewer #1 (Remarks to the Author):

Barskiy et al extend the use of an emerging magnetometer approach, ZULF NMR, to the characterization of rapid exchange processes in this study using ammonium as a model system. They further implicate the wide spread utilization of this approach by measuring hyperpolarized [2-13C] pyruvic acid. While dissolution DNP is used as the method for hyperpolarization, it is of course not required.

The studies of ammonium and the pH dependent changes in the ZULF spectrum are well thought out, matching simulation to data. However, the extension to dDNP with HP [2-13C] pyruvic acid requires further data to demonstrate that it could be useful in the biochemical contexts the authors claim and furthermore broad applicability:

- (1) For example, a spectrum of lactate is simulated, this is hardly a difficult experiment for the investigators to accomplish. At the very least such an in vitro experiment should be explored.
- (2) What is the role of T1-relaxation in the setting of ZULF NMR and why is the achieved SNR so minuscule given such a dramatically hyperpolarized sample (polarization = 15%)? Would this approach then be relevant in the context of basic benchtop permanent magnet style detection?
- (3) The authors show a single measurement per sample. Does the Zero field or ultra-low field play a substantial role in the evolution of hyperpolarized spins? Is it possible to re-measure an evolving sample? This would be required for such an application to a non-equilibrium, evolving system.

Reviewer #2 (Remarks to the Author):

The authors present a novel application of zero to ultra-low field NMR to the study of chemical exchange of small molecules in solution. They record spectra of ammonium ions at different pH concentrations at high field as well as at zero field to investigate the effects of exchange rate on NMR line shapes. As increasing the pH, the data show characteristic multiplet pattern coalescence in the first case and vanishing of the resonances in the second. Numerical simulations of the spectra including relaxation allow quantifying the rate of the proton exchange in most cases. Using Markov's chain calculation, the authors explain how the ZULF signals vanish with pH increase by the fact that some of the spin order is lost during the shuttling from the prepolarizing magnets. In that, they extend the knowledge of which systems are suitable or not for ZULF spectroscopy.

In addition to that, the authors perform an experiment where ZULF is coupled to dissolution dynamic polarization on pyruvic acid, the paragon of dDNP, and they record its ZULF resonances.

The science presented in this manuscript is of high clarity and we recommend acceptance of the manuscript with minor revision once the following points have been addressed:

Comments:

- References. many references numbers are incorrect.
- Abstract. While chemical shifts...other. One could argue that his argument is irrelevant, as at high field chemical shifts (in Hz) often spread more than J (in Hz).
- P 4. Which are achieved... NMR can also be performed in moderate fields in transportable systems, though with potentially less sensitivity and/or resolution.
- P 4. with a high sensitivity, without need for strong, persistent magnetic fields. A high sensitivity if compared to what? For example how does sensitivity compare with NMR at 500 MHz?
- P 5. Lose. typo
- P 9. According to the method part, the studied solutions were not degassed. Can the paramagnetic relaxation via dissolved oxygen be neglected in the incoherent relaxation?
- P 9. the spin state of B (ρ_B) is assumed to be known and its value is kept constant.

Constant but equal to what exactly? Can the authors be more specific?

- P 10. The state of freely exchanging protons in solution can be considered uncorrelated and unpolarized. What about when a proton jumps from the acid to the conjugate base? Is it neglectable? Should it be considered uncorrelated and unpolarized in this case?

- P 10. iRelax...elsewhere. We recommend that at least the minimum information needed to reproduce these simulations be given in supplement.
- P 14. The liquid ... dissolution-DNP instrument.

Can the authors give more details on the DNP parameters, such as microwave frequency and power? Which nuclear spins are polarized? Probably ^{13}C but potentially also ^1H ? Was it positive or negative polarization? Were the final polarization levels measured in the solid state?

- P 14. The total time delay between dissolution of the hyperpolarized material and acquisition of the first zero-field NMR spectrum was ~ 15 s.

Several scans are mentioned. So a decay curve must be available. Why is not shown (at least in SI)?

- P 19. thus accelerating ZULF NMR signal acquisition by a factor of $\sim 10,000$ (hundreds of scans would have been needed to detect a signal with the same SN).

An enhancement of 10'000 would correspond to $10'000^2 = 1\text{E}8$ scans, rather than 100s of scans, as the authors state. A better description and evaluation of the enhancement factor would be important here.

- Figure 3a. The authors should give enhancement factor with respect to the thermal equilibrium experiment with shuttling from the permanent magnet. This enhancement could at least be calculated by comparing with a highly concentrated reference solution.
- Figure 3c. From the text and the absence of noise, it is clear that these spectra are simulations but please specify it for clarity.
- P 22. did not give different FWHHs for the resulting peaks even when using different dissociation-association exchange rates

This sentence is unclear. On Figure S9, one can see the peaks broadening and then coalescing into two peaks. What does the authors mean by saying that "numerical simulations [...] did not give different FWHHs for the resulting peaks". Different to what then?

- P 24. the same fact can be advantageous when the studied system is complex (e.g., cell cultures or bioreactors) and observation of only selected chemical pathways is desired. Indeed the loss of signal would remove resonances and simplify spectra but in an uncontrolled manner. Is it really advantageous?
- P 26. reactors with complicated geometry

It is not obvious to see how ZULF can allow to study reactors with complicated geometry when the sample needs to be shuttled in and out of the shield. I suppose the authors mean that ZULF may be relevant in this context when coupled with hyperpolarization techniques. This deserves to be specified.

- P 27. Based on simulations, conversion of pyruvate to lactate

Rather: "Based on simulations, we demonstrate that conversion of pyruvate to lactate" or similar

Reviewers' comments:

Reviewer #1 (Remarks to the Author):

Barskiy et al extend the use of an emerging magnetometer approach, ZULF NMR, to the characterization of rapid exchange processes in this study using ammonium as a model system. They further implicate the wide spread utilization of this approach by measuring hyperpolarized [2-¹³C] pyruvic acid. While dissolution DNP is used as the method for hyperpolarization, it is of course not required.

The studies of ammonium and the pH dependent changes in the ZULF spectrum are well thought out, matching simulation to data.

Authors' reply: We thank the referee for his/her assessment of the studies as "well thought out".

Changes to the manuscript: None.

However, the extension to dDNP with HP [2-¹³C] pyruvic acid requires further data to demonstrate that it could be useful in the biochemical contexts the authors claim and furthermore broad applicability:

(1) For example, a spectrum of lactate is simulated, this is hardly a difficult experiment for the investigators to accomplish. At the very least such an *in vitro* experiment should be explored.

Authors' reply: We thank the referee for his/her comment. We agree that biochemical conversion of pyruvate to lactate monitored by zero-field NMR could be a ground-breaking contribution to the field of hyperpolarization. Such a study is certainly warranted and currently we are working on constructing a new device to perform the *in vitro* and *in vivo* measurements. The main focus of the current paper is to demonstrate the importance of chemical exchange phenomena on ZULF NMR spectra as well as to show the exemplary experiment relevant to the field of biomedicine (and not to perform in-depth metabolic studies by hyperpolarized ZULF NMR).

Changes to the manuscript: We changed multiple sentences in the manuscript to make the main point clear and to give less emphasis to the detection of pyruvate-lactate conversion via ZULF NMR. The modified sentences are:

Old sentence: "Given these findings, we suggest possibilities of ZULF NMR coupled with hyperpolarization as a valuable tool for the analysis of biochemically-relevant chemically exchanging systems in places where conventional, high-field NMR is hard to implement."

New sentence: "Given these findings, we suggest ways to detect hyperpolarized molecules by portable ZULF NMR spectrometers in places where conventional, high-field NMR is hard to implement."

Old caption of the Figure 3c: "Biochemical conversion of pyruvate to lactate is an important target for studies of metabolism *in vivo*. As shown in this simulated ZULF NMR spectrum, it is possible to detect pyruvate→lactate conversion with zero-field NMR due to narrow (sub-Hz) resonances separated by more than 100 Hz, a typical carbon-proton *J*-coupling value. Note that in the simulation, we did not consider relaxation/decoherence effects."

New caption of the Figure 3c: "Simulated ZULF NMR spectrum of [2-¹³C]pyruvate and [2-¹³C]lactate. Biochemical conversion of pyruvate to lactate is an important target for studies of

metabolism *in vivo*. ZULF NMR detection of hyperpolarized molecules may endow narrow (sub-Hz) resonances separated by more than 100 Hz, a typical carbon-proton J -coupling value. Note that in the simulation, we did not consider relaxation/decoherence effects.”

Old sentence: “Practically relevant applications of ZULF NMR may include detection of microscopic biological samples such as cell cultures⁵¹ and reactors with complicated geometry (which are hard to study by conventional NMR due to their size).”

New sentence: “Practically relevant applications of ZULF NMR if combined with any of the available hyperpolarization techniques may include detection of microscopic biological samples such as cell cultures and reactors with complicated geometry (which are hard to study by conventional NMR due to their size).”

Old sentence: “Based on simulations, conversion of pyruvate to lactate can be monitored via ZULF NMR under biological conditions given the large frequency difference between the corresponding J -spectra.”

New sentence: “Based on simulations, we speculate that conversion of pyruvate to lactate can be monitored via ZULF NMR under biological conditions given the large frequency difference between the corresponding J -spectra.”

(2) What is the role of T_1 -relaxation in the setting of ZULF NMR and why is the achieved SNR so minuscule given such a dramatically hyperpolarized sample (polarization = 15%)? Would this approach then be relevant in the context of basic benchtop permanent magnet style detection?

Authors' reply: We thank the referee for his/her comment. As noted in the manuscript:

“The sensitivity of the prototype portable spectrometer was estimated to be in the range of 500-1000 fT/Hz^{1/2} which is two orders of magnitude worse than the recently achieved sensitivity of 10-20 fT/Hz^{1/2} on similar devices”.

More than two orders of magnitude enhancements in SNR are expected if the sensitivity is extended to this practically achievable level of 10 fT/Hz^{1/2}. We also added a figure (Fig. S2) to the supporting information to describe the role of T_1 relaxation on the signal of the hyperpolarized sample. Significant improvement in detection sensitivity is expected for an optimized ZULF NMR device with integrated fluid transfer path, which would minimize relaxation losses during the transfer from the polarizer. For these reasons, we have no doubts that optimized setup would be a useful tool to study biochemical transformations *in vitro* and *in vivo*. We also agree that basic benchtop detection experiments are warranted and such studies are currently under way.

Changes to the manuscript: additional figure (Fig. S2) describing the role of T_1 relaxation is added to the supporting information.

(3) The authors show a single measurement per sample. Does the Zero field or ultra-low field play a substantial role in the evolution of hyperpolarized spins? Is it possible to re-measure an evolving sample? This would be required for such an application to a non-equilibrium, evolving system.

Authors' reply: We thank the referee for his/her comment. Decoherence at ultralow magnetic field is a significant but poorly understood problem for many substrates hyperpolarized by dissolution DNP. Substantial losses of polarization can occur during transport between dissolution DNP and NMR/MRI apparatus as the result of near-zero-field crossings in the laboratory magnetic field, plus paramagnetic relaxation due to radicals and chemical

exchange. Obviously, the motivation for hyperpolarized NMR is to increase signal strength; any losses between hyperpolarization and detection of the nuclear spins must be minimized.

Changes to the manuscript: additional figure (Fig. S2) was added to the supporting information to show ZULF NMR spectra of four successive measurements after the sample storage at Earth's magnetic field.

Reviewer #2 (Remarks to the Author):

The authors present a novel application of zero to ultra-low field NMR to the study of chemical exchange of small molecules in solution. They record spectra of ammonium ions at different pH concentrations at high field as well as at zero field to investigate the effects of exchange rate on NMR line shapes. As increasing the pH, the data show characteristic multiplet pattern coalescence in the first case and vanishing of the resonances in the second. Numerical simulations of the spectra including relaxation allow quantifying the rate of the proton exchange in most cases. Using Markov's chain calculation, the authors explain how the ZULF signals vanish with pH increase by the fact that some of the spin order is lost during the shuttling from the prepolarizing magnets. In that, they extend the knowledge of which systems are suitable or not for ZULF spectroscopy.

In addition to that, the authors perform an experiment where ZULF is coupled to dissolution dynamic polarization on pyruvic acid, the paragon of dDNP, and they record its ZULF resonances.

The science presented in this manuscript is of high clarity and we recommend acceptance of the manuscript with minor revision once the following points have been addressed:

Authors' reply: We thank the referee for his/her assessment of the science in the manuscript as "of high clarity" and recommending the acceptance of the manuscript.

Changes to the manuscript: None.

Comments:

- References. many references numbers are incorrect.

Authors' reply: We thank the referee for the comment, we agree.

Changes to the manuscript: References were carefully checked and errors were eliminated.

- Abstract. While chemical shifts...other. One could argue that his argument is irrelevant, as at high field chemical shifts (in Hz) often spread more than J (in Hz).

Authors' reply: We thank the referee for the comment. We agree.

Changes to the manuscript: The sentence was removed from the abstract per reviewer's suggestion.

- P 4. Which are achieved... NMR can also be performed in moderate fields in transportable systems, though with potentially less sensitivity and/or resolution.

Authors' reply: We thank the referee for the comment, we agree.

Changes to the manuscript: The sentence was rewritten. The new paragraph reads:

Despite the tremendous utility of conventional high-field NMR, it has several limitations. First, it requires high magnetic fields (typically several Tesla, often achieved by using superconducting magnets) and, therefore, is costly and hard to transport. Second, cryogenics needed for operation of an NMR magnet necessitate advanced research infrastructure. While NMR can also be performed in transportable permanent magnet systems (1-2 T) though with reduced sensitivity and resolution, it is also possible to study the behavior of nuclear spins in the complete absence of an applied field.

- P 4. with a high sensitivity, without need for strong, persistent magnetic fields. A high sensitivity if compared to what? For example how does sensitivity compare with NMR at 500 MHz?

Authors' reply: We thank the referee for pointing to the misstep. We added an actual number specifying the high sensitivity (as well as the corresponding reference). We note that a major advantage of ZULF NMR (compared to high-field NMR) for studying liquid samples is the high resolution (narrow spectral line widths) resulting from absence of magnetic field gradients and long natural coherence times.

Changes to the manuscript: the sentence was modified:

This form of NMR, also known as “zero- or ultralow-field” (ZULF) NMR, has the ability to provide chemical resolution in mixtures⁹⁻¹⁰ (narrow spectral lines arising from absence of magnetic-field gradients and long coherence times) with a high sensitivity (10 fT/Hz^{0.5}), without need for strong, persistent magnetic fields.¹¹⁻¹²

- P 5. Lose. Typo

Authors' reply: We modified the sentence.

Changes to the manuscript: The sentence now reads:

We additionally rationalize these findings by the Markov-chain analysis of “the nuclear spin memory” in the system and demonstrate that only ~10 proton dissociation-association events in $^{15}\text{NH}_4^+$ are enough to lose nuclear spin correlations necessary for observation of the ZULF NMR spectrum.

- P 9. According to the method part, the studied solutions were not degassed. Can the paramagnetic relaxation via dissolved oxygen be neglected in the incoherent relaxation?

Authors' reply: We thank the referee for this important comment. From the experimental standpoint, concentration of oxygen was the same in all samples and therefore it was not considered in simulations. When discussing the reasons of differential line broadening at zero field, we touch upon this topic:

“Ongoing chemical exchange (hydration) and field gradients across the sample both lead to “smoothing” of the spectral features and may explain differential line broadening. We note that the presence of paramagnetic impurities (e.g., not completely filtered OX063 radicals used for dDNP process) can also contribute to the observed phenomenon.”

Changes to the manuscript: None.

- P 9. the spin state of B (ρ_B) is assumed to be known and its value is kept constant. Constant but equal to what exactly? Can the authors be more specific?

Authors' reply: We thank the referee for the comment. In this section we added a little of context (see changes). However, more details for practical cases are in fact given in the manuscript. The text follows as

“Convenience of the set of equations (3) lies in the fact that the state of freely exchanging protons in solution can be considered uncorrelated and unpolarized, i.e., $\hat{\rho}_B = \hat{\mathbf{1}}_B / \text{Tr} \{ \hat{\mathbf{1}}_B \}$ (here $\hat{\mathbf{1}}_B$ is the unit matrix) and, therefore, embedded into the matrix \hat{M} . This is reasonable since T_1/T_2 relaxation of exchanging protons [H⁺] in aqueous solutions is known to be fast on the relevant timescales (see below).²⁴”

Changes to the manuscript: We added more description to the text per reviewer's suggestion:

In our approach, $\hat{\rho}_A$ and $\hat{\rho}_C$ are columns produced from corresponding square matrices by column-wise concatenation and the spin state of B ($\hat{\rho}_B$) depends on the experiment and its value is kept constant. For example, an ensemble of protons can be considered completely unpolarized at zero magnetic field giving $\hat{\rho}_B = \hat{\mathbf{1}}_B / \text{Tr} \{ \hat{\mathbf{1}}_B \}$ (see below).

- P 10. The state of freely exchanging protons in solution can be considered uncorrelated and unpolarized. What about when a proton jumps from the acid to the conjugate base? Is it neglectable? Should it be considered uncorrelated and unpolarized in this case?

Authors' reply: We thank the referee for the comment. Given concentrations of the materials studied, the main contribution of the exchange is intermolecular jumps between the proton and a solvent. Proton jumps between the acid and corresponding conjugate base are also possible but they are significantly less probable. Indeed, in case of ammonium for the pH levels studied in this work, the concentration of the conjugate base, NH₃, varies from 0.0001% to 0.001% of 6 M (see Figure 2b), compared to the water concentration of 55.6 M. Therefore, it has a miniscule effect on the observed dynamics.

Changes to the manuscript: None.

- P 10. iRelax...elsewhere. We recommend that at least the minimum information needed to reproduce these simulations be given in supplement.

Authors' reply: We thank the referee for his/her comment. These simulations can be reproduced using the open-source MOIN spin library. All used scripts are available online.

Changes to the manuscript: The following sentence was added

Simulations were performed using the MOIN spin library (available online)²⁹ that has been shown to efficiently model nuclear spin dynamics in chemically exchanging systems.³⁰

- P 14. The liquid ... dissolution-DNP instrument.
Can the authors give more details on the DNP parameters, such as microwave frequency and power? Which nuclear spins are polarized? Probably ¹³C but potentially also ¹H? Was it positive or negative polarization? Were the final polarization levels measured in the solid state?

Authors' reply: We thank the referee for the comment.

Changes to the manuscript: We modified the method section to include additional information per reviewer's suggestion:

“The liquid was frozen into a glass, then polarized for 45 minutes at 1.3 K in a commercial dissolution-DNP instrument (3.35 T, Oxford Instruments HyperSense, Abingdon, UK), following a standard protocol (microwave frequency: 94.099 GHz; microwave power: 25mW; polarized on the positive lobe of EPR line).³¹ Since the narrow-line radical OX063 was used in the sample preparation, we expect mainly ¹³C to be polarized via solid effect. Polarization level of ~15% was estimated from the amplitude of the solid-state ¹³C NMR signal prior to dissolution.” Comparison of the ¹³C NMR signal intensity between the hyperpolarized sample and thermally polarized sample was done while the sample was inside the Hypersense instrument.

- P 14. The total time delay between dissolution of the hyperpolarized material and acquisition of the first zero-field NMR spectrum was ~15 s. Several scans are mentioned. So a decay curve must be available. Why is not shown (at least in SI)?

Authors' reply: We thank the referee for the comment.

Changes to the manuscript: an additional figure (Fig. S2) was added to the supporting information to demonstrate the decay curve.

- P 19. thus accelerating ZULF NMR signal acquisition by a factor of ~10,000 (hundreds of scans would have been needed to detect a signal with the same SN. An enhancement of 10'000 would correspond to $10'000^2=1E8$ scans, rather than 100s of scans, as the authors state. A better description and evaluation of the enhancement factor would be important here.

Authors' reply: We thank the referee for the comment. By saying that signal acquisition would be enhanced (i.e., accelerated) by the factor of 10,000 we meant decreasing the time necessary to acquire the same SNR. The confusion probably comes from the fact that in order to increase SNR by the factor of N, N² acquisitions should be taken. We did not talk about the enhancement in SNR.

Changes to the manuscript: None.

- Figure 3a. The authors should give enhancement factor with respect to the thermal equilibrium experiment with shuttling from the permanent magnet. This enhancement could at least be calculated by comparing with a highly concentrated reference solution.

Authors' reply: We agree. We did the measurement and analyzed the enhancement.

Changes to the manuscript: an additional section (2) was added to the supporting information.

- Figure 3c. From the text and the absence of noise, it is clear that these spectra are simulations but please specify it for clarity.

Authors' reply: We agree. We modified the caption of the Figure 3 to highlight the fact that the spectrum is a simulation.

Changes to the manuscript: The figure caption was modified per referee's suggestion.

- P 22. did not give different FWHHs for the resulting peaks even when using different dissociation-association exchange rates
This sentence is unclear. On Figure S9, one can see the peaks broadening and then

coalescing into two peaks. What does the authors mean by saying that "numerical simulations [...] did not give different FWHHs for the resulting peaks". Different to what then?

Authors' reply: We meant that upon broadening, both peaks have the same resulting FWHH. We agree that the sentence was not clear.

Changes to the manuscript: The text was rewritten to avoid the confusion:

"However, carrying out numerical simulations of ZULF NMR spectra for these molar fractions of hydrated and non-hydrated forms and assuming chemical exchange as a $(A_3X)B_2 \rightleftharpoons A_3X + 2B$ process [here $(A_3X)B_2$ is 2,2-dihydroxypropionic acid, A_3X is pyruvic acid and 2B are protons of water] did not give different FWHHs for the resulting peaks upon exchange-induced broadening (Figure S10)."

- P 24. the same fact can be advantageous when the studied system is complex (e.g., cell cultures or bioreactors) and observation of only selected chemical pathways is desired. Indeed the loss of signal would remove resonances and simplify spectra but in an uncontrolled manner. Is it really advantageous?

Authors' reply: We thank the referee for the comment. As an advantage, we meant the absence of the J -coupled spectra from solvents such as water. In conventional, high-field NMR, one needs to use deuterated solvents to remove the large background signal. This is unnecessary for ZULF NMR and non-labelled, protonated solvents can be used. This is particularly important for studies of biological samples.

Changes to the manuscript: The text was rewritten

"However, the same fact can be advantageous for eliminating the signal from solvents such as water or when the studied system is complex (e.g., cell cultures or bioreactors) and observation of only selected chemical pathways is desired."

- P 26. reactors with complicated geometry
It is not obvious to see how ZULF can allow to study reactors with complicated geometry when the sample needs to be shuttled in and out of the shield. I suppose the authors mean that ZULF may be relevant in this context when coupled with hyperpolarization techniques. This deserves to be specified.

Authors' reply: We thank the referee for the comment. We agree.

Changes to the manuscript: The text was modified per reviewer's suggestion:

"Practically relevant applications of ZULF NMR if combined with any of the available hyperpolarization techniques may include detection of microscopic biological samples such as cell cultures⁵¹ and reactors with complicated geometry (which are hard to study by conventional NMR due to their size)."

- P 27. Based on simulations, conversion of pyruvate to lactate
Rather: "Based on simulations, we demonstrate that conversion of pyruvate to lactate" or similar

Authors' reply: We agree.

Changes to the manuscript: The text was modified per reviewer's suggestion.

REVIEWERS' COMMENTS:

Reviewer #1 (Remarks to the Author):

The authors have addressed my previous critiques.

Reviewer #2 (Remarks to the Author):

The manuscript has been greatly improved and we recommend acceptance after the following minor question has been addressed. Figure S2 shows relaxation of hyperpolarized pyruvate and is fitted with a linear regression. Could the authors comment on that, and possibly change the fit to a monoexponential decay ?

Response to referees

Reviewers' comments:

Reviewer #1 (Remarks to the Author):

The authors have addressed my previous critiques.

Changes to the manuscript: None.

Reviewer #2 (Remarks to the Author):

The manuscript has been greatly improved and we recommend acceptance after the following minor question has been addressed. Figure S2 shows relaxation of hyperpolarized pyruvate and is fitted with a linear regression. Could the authors comment on that, and possibly change the fit to a monoexponential decay?

Authors' reply: We thank the referee for the question. We agree and change the fit to monoexponential decay.

Changes to the manuscript: Supplementary Figure 2 was modified per reviewer's suggestion.